

# Agricultural land-use change in a Mexican oligotrophic desert depletes ecosystem stability

Natali Hernández-Becerra[1], Yunuen Tapia-Torres[2], Ofelia Beltrán-Paz[1], Jazmín Blaz[3], Valeria Souza[3] and Felipe García-Oliva[1]

[1] Laboratorio de biogeoquímica de suelos, Instituto de Investigaciones en Ecosistemas y Sustentabilidad, UNAM, Morelia, Michoacán, Mexico
[2] ENES Unidad Morelia, Universidad Nacional Autónoma de México, Morelia, Michoacán, Mexico
[3] Instituto de Ecología, Universidad Nacional Autónoma de México, Mexico

Corresponding author
Felipe García-Oliva,
fgarcia@cieco.unam.mx

## ABSTRACT

**Background**. Global demand for food has led to increased land-use change, particularly in dry land ecosystems, which has caused several environmental problems due to the soil degradation. In the Cuatro Cienegas Basin (CCB), alfalfa production irrigated by flooding impacts strongly on the soil.

**Methods**. In order to analyze the effect of such agricultural land-use change on soil nutrient dynamics and soil bacterial community composition, this work examined an agricultural gradient within the CCB which was comprised of a native desert grassland, a plot currently cultivated with alfalfa and a former agricultural field that had been abandoned for over 30 years. For each site, we analyzed C, N and P dynamic fractions, the activity of the enzyme phosphatase and the bacterial composition obtained using 16S rRNA clone libraries.

**Results**. The results showed that the cultivated site presented a greater availability of water and dissolved organic carbon, these conditions promoted mineralization processes mediated by heterotrophic microorganisms, while the abandoned land was limited by water and dissolved organic nitrogen. The low amount of dissolved organic matter promoted nitrification, which is mediated by autotrophic microorganisms. The microbial N immobilization process and specific phosphatase activity were both favored in the native grassland. As expected, differences in bacterial taxonomical composition were observed among sites. The abandoned site exhibited similar compositions than native grassland, while the cultivated site differed.

**Discussion**. The results suggest that the transformation of native grassland into agricultural land induces drastic changes in soil nutrient dynamics as well as in the bacterial community. However, with the absence of agricultural practices, some of the soil characteristics analyzed slowly recovers their natural state.

## INTRODUCTION

Rising global food demand due to population growth has caused an increase in rates of land-use change to agricultural production in dry ecosystems (*Lepers et al., 2005*; *Reynolds et*

*al., 2007*). This has led to several environmental problems, including deforestation, habitat fragmentation, biodiversity reduction, changes to global biogeochemical cycles, water and soil contamination and degradation (*Reynolds et al., 2007*; *Rey-Benayas & Bullock, 2012*). In these perturbed dry lands, the main drivers of desertification are soil nutrient losses caused mainly by erosion, soil salinization and the reduction of soil water retention capacity through the deterioration of soil physical properties (*D'Odorico et al., 2013*). This soil degradation reduces agricultural productivity and the fields are eventually abandoned.

The main characteristics of intensive agriculture that affect soil properties is the reduction of organic matter inputs, soil tillage, fertilization and irrigation (*McLauchlan, 2006*). It has been reported that soil organic matter (SOM) is reduced by 16–77%, as a consequence of agriculture (*Murty et al., 2002*). This is mainly through the decrease in organic matter inputs and the increase in soil organic decomposition because of increased tillage and soil temperatures (*Trasar-Cepeda et al., 2008*; *Beheshti, Raiesi & Golchin, 2012*). The practice of tillage disrupts the physical properties of the soil, affecting soil water and nutrient dynamics (*Six, Elliott & Paustian, 1999*; *Zeleke et al., 2004*; *Bronick & Lal, 2005*). Fertilization with nitrogen, mainly in the form of ammonium, promotes faster nitrification and the release of $H^+$ ions into the soil solution, thus lowering soil pH (*Moore, Klose & Tabatabai, 2000*) and the continuous irrigation increases the leaching of salts through the soil profile (*Raiesi, 2004*). However, when agricultural fields are abandoned, some salts accumulate in the topsoil, promoting salinization, a process that is favored in desert ecosystems (*Rietz & Haynes, 2003*; *Pan et al., 2012*). Furthermore, plant succession is slower in desert ecosystems than in wet tropical ecosystems; for example, recovery of vegetation requires at least 40 years in the former, while in the latter it can be achieved in less than 10 years (*Lesschen et al., 2008*; *Wang et al., 2011*).

Agriculture also has an effect on the composition of the soil microbial community. For instance, some changes in microbial composition have been reported as a result of agricultural land-use in tropical (*Waldrop, Balser & Firestone, 2000*) as well as desert (*Ding et al., 2013*) and Mediterranean ecosystems (*Garcia-Orenes et al., 2013*). However, the effect on soil microbial diversity is unclear; some studies have described increases in biodiversity (*Jangid et al., 2008*) while others have reported decreases (*Lupwayi, Rice & Clayton, 1998*). *Chaudhry et al. (2012)* found higher soil microbial diversity in agricultural fields managed with organic rather than chemical fertilization. These authors found that the composition of the bacterial community in the organically fertilized soil was dominated by the phyla Proteobacteria, Bacteroidetes and Gemmatimonadetes, while the groups Actinobacteria and Acidobacteria were predominant in the chemically fertilized soil. The dominant phyla in the organically fertilized soil have been associated with high nutrient availability, whereas the Acidobacteria have been related to nutrient-poor soils (*Fierer, Bradford & Jackson, 2007*). The effect of long-term agricultural management on soil microbial communities is similarly unclear; in some studies, even after 9 years of abandonment, the soil microbial composition remains similar to that of the cultivated soil (*Buckley & Schmidt, 2001*). However, an agricultural field abandoned for over 45 years presented a soil microbial community that was similar to one in soil with native vegetation cover (*Buckle & Schmidt, 2003*). These results demonstrate the need for further study in order

to understand the effect of succession of agriculture management upon the composition of the soil microbial community.

The worldwide area of degraded agriculture fields was estimated to be 12,400,000 km$^2$ in 2007 (*Rey-Benayas & Bullock, 2012*), of which 20% corresponded to dry ecosystems (*Lepers et al., 2005*; *Reynolds et al., 2007*). In Mexico, around 121 km$^2$ and 45 km$^2$ of grassland were converted to agriculture and abandoned lands, respectively, between 2005 and 2010 (*Colditz, Llamas & Ressl, 2014*). For this reason, evaluation of the capacity for soil restoration in the cultivated fields of dry lands is a priority for crop production and ecosystem conservation. This capacity can be evaluated in the context of ecosystem stability, which has two main components: resistance and resilience (*Pimm, 1984*). The former is the capacity of the ecosystem to face a disturbance without undergoing structural changes, while the latter reflects the time required for the ecosystem to return to its pre-disturbance condition (*Pimm, 1984*). *Orwin & Wardle (2004)* proposed indices for evaluating these two attributes of soil stability, which are accurate for providing a relative quantitative measurement when comparing soil conditions under perturbation. The quantitative measure of soil stability allows evaluation of the magnitude of soil degradation and its capability for restoration.

In the Cuatro Cienegas basin (CCB) in Mexico, alfalfa (*Medicago sativa* L.) production with gravity irrigation involves flooding the fields with oasis water that is channeled through open canals for hundreds of km. This practice unequivocally threatens the sustainability of the CCB wetland and degrades the soil and vegetation. In order to analyze the effect of such agricultural land-use on the soil nutrient dynamics (C, N and P) and composition of the soil bacterial community, we examined an agricultural gradient within the CCB composed of three sites with the same soil type but under contrasting management: a native desert grassland, a plot with an alfalfa crop and a former agricultural field that had been abandoned for over 30 years. We predicted that the alfalfa production disrupts the mechanisms of soil nutrient transformation and strongly affects the composition of the soil bacteria. To test these hypotheses, we analyzed C, N and P dynamic fractions and used this data to calculate the homeostasis of the microbial community. The enzymatic activity of alkaline phosphatase was also quantified and bacterial composition was determined through the use of 16S rRNA clone libraries.

# MATERIAL AND METHODS

## Site description

This study was carried out in the Cuatro Cienegas basin (CCB; 26°50′N and 102°8′W) at 740 masl, in the Chihuahuan desert in Mexico. The climate is seasonally arid with an average annual temperature of 21 °C and annual precipitation of 252 mm (http://smn.cna.gob.mx/). Jurassic-era gypsum is the dominant parent material on the western side of the basin, while Jurassic-era limestone dominates on the eastern side (*McKee, Jones & Long, 1990*). According to the WRB classification (2007), the predominant soils are Gypsisol and Calcisol on the western and eastern sides of the basin, respectively. The soil within the CCB is characterized by low P concentrations (ranging between 70–200 μg g$^{-1}$).

These values are lower than the $P$ values of other soils within the Chihuahuan desert (500–1,000 $\mu$g g$^{-1}$; *Tapia-Torres & Garcia-Oliva, 2013*). The main vegetation types are halophyte-grassland dominated by *Sporobolus airoides* (Poaceae) and desert scrub dominated by species from the Euphorbiaceae and Zygophyllaceae families (*Perroni, Garcia-Oliva & Souza, 2014*). Agricultural activity in the CCB began in the early decades of the 20th century but has increased in the last 30 years and it mainly consists of the production of alfalfa for cattle fodder. Alfalfa (*Medicago sativa* L.) is grown by flooding the fields and introducing large quantities of fertilizer. In some years, sorghum (*Sorghum* spp.) is cultivated, but the alfalfa cultivation dominates the agricultural surface (*INEGI, 2011*). However, these fields must eventually be abandoned due to degradation of the soil, mainly through salinization.

## Field sampling

Sampling sites were located on the eastern side of the CCB. An agricultural gradient was established comprising three sites of shared soil type (Calcisol) but contrasting management was all located in flat areas: native desert grassland, a plot cultivated with alfalfa and a former agricultural field that had been abandoned for over 30 years. The native desert grassland was in the Pozas Azules reserve (26°49′30″N and 102°1′27″W) where *Sporobolus airoides* is the dominant plant species (*Tapia-Torres et al., 2015a*). The cultivated alfalfa field was located in the Cuatro Cienegas ejido (26°58′47″N and 102°02′13″W) and covered an area of 2.7 ha with high fertilizer inputs and irrigation by flooding every month. The plot was fertilized with monoammonium phosphate (11-52-00) dissolved in the water for irrigation. The water for irrigation had a pH value of 8.5 with a high electrical conductivity (150 mS m$^{-1}$). This alfalfa plot has been under cultivation for 20 consecutive years and the alfalfa is harvested every month. Finally, the abandoned field was also in the Cuatro Cienegas ejido (26°58′57″N and 102°01′8″W) and presented minimum plant cover (less than 30% of the area). Oscar Sánchez Liceaga, Héctor Castillo González, the personnel of APFF Cuatro Cienegas (CONANP) and the people in charge of Rancho Pozas Azules (PRONATURA) gave us the permission to collect soil samples on their respective properties. At each site, a $100 \times 50$ m plot was delimited and then divided into 10 sections at a distance of 10 m apart. A random sampling transect was then established in each section, with topsoil samples taken to a depth of 15 cm at ten sampling points (every five meters) in September 2011; these samples were then mixed to form one composite sample. In total, 10 such composite samples were taken in each plot. Soil for biogeochemical and enzymatic activity analysis was stored in black plastic bags and refrigerated at 4 °C. In order to characterize the bacterial community at each site, 100 g of composite samples were immediately stored in liquid nitrogen until subsequent DNA extraction.

## Laboratory analyses
### Soil nutrient and enzymatic analyses

Soil pH was measured in deionized water (1:2 w:v) using a digital pH meter (Corning) and soil electrical conductivity was measured by conductivity meter (Hannan Instruments Inc., Houston, USA). A subsample (100 g) was oven-dried at 75 °C to constant weight for soil moisture determination using the gravimetric method in order to adjust for water

content when expressing nutrient concentration on the basis of dry soil mass. All C forms analyzed in all samples were determined in a total carbon analyzer (UIC model CM5012, Chicago, USA), while the N and P forms analyzed were determined colorimetrically in a Bran-Luebbe Auto analyzer 3 (Norderstedt, Germany). Prior to the total soil nutrient analyses, soil samples were dried and ground with a pestle and mortar. Total carbon (TC) and inorganic carbon (IC) were determined by combustion and coulometric detection (*Huffman, 1977*). Total organic carbon (OC) was calculated as the difference between TC and IC. For total N (TN) and total P (TP) determination, samples were acid digested with $H_2SO_4$, $H_2O_2$, $K_2SO_4$ and $CuSO_4$ at 360 °C. Soil N was determined by the macro-Kjeldahl method (*Bremmer, 1996*), while P was determined by the molybdate colorimetric method following ascorbic acid reduction (*Murphy & Riley, 1962*).

Available, dissolved and microbial nutrient forms were extracted from field moist soil samples. Available inorganic N ($NH_4^+$ and $NO_3^-$) was extracted from 10 g of fresh soil subsamples with 2M KCl, followed by filtration through a Whatman No. 1 paper filter (*Robertson et al., 1999*) and determined colorimetrically by the phenol-hypochlorite method. Available (inorganic) and labile (organic) P was determined by extraction with 0.5M $NaHCO_3$ at pH 8.5 according to Hedley sequential P fractionation (*Tiessen & Moir, 1993*) and quantified as described above for orthophosphate.

Dissolved nutrients were extracted with deionized water after shaking for 45 min and filtering through a Millipore 0. 42 00B5m filter (*Jones & Willett, 2006*). Prior to acid digestion, one aliquot of the filtrate was used to determine dissolved ammonium ($DNH_4^+$) and inorganic P (IP) in deionized water extract. Total dissolved nitrogen (TDN) was digested using the macro-Kjeldahl method. Total dissolved P (TDP) was also acid digested and determined by colorimetry. Total dissolved carbon (TDC) was measured with an Auto Analyzer of carbon (TOC CM 5012) module for liquids (UIC-COULOMETRICS). Inorganic dissolved carbon (IDC) was determined in an acidification module CM5130. Dissolved organic carbon (DOC), dissolved organic nitrogen (DON) and dissolved organic phosphorous (DOP) were calculated as the difference between total dissolved forms and inorganic dissolved forms.

Microbial C ($C_{mic}$), N ($N_{mic}$) and P ($P_{mic}$) concentrations were determined by the chloroform fumigation extraction method (*Vance, Brookes & Jenkinson, 1987*). Fumigated and non-fumigated samples were incubated for 24 h at 25 °C and constant moisture. Microbial C was extracted from fumigated and non-fumigated samples with 0.5 M $K_2SO_4$ and filtered through Whatman No. 42 filters (*Brookes et al., 1985*). The concentration of C was measured in each extract as total and inorganic C concentration by the method described before. Microbial C was calculated by subtracting the extracted carbon in non-fumigated samples from that of fumigated samples and dividing the result by a $K_{EC}$ value (the extractable part of microbial biomass C) of 0.45 (*Joergensen, 1996*). Microbial N was extracted with the same procedure used for $C_{mic}$, but the extract was filtered through Whatman No. 1 paper. The filtrate was acid digested and determined as TN by Macro-Kjeldahl method (*Brookes et al., 1985*). Microbial N was calculated as for $C_{mic}$, but divided by a $K_{EN}$ value (the extractable part of microbial biomass N after fumigation) of 0.54 (*Joergensen & Mueller, 1996*). Microbial P was extracted using $NaCO_3$ 0.5M at pH

8.5, after which the fumigation-extraction technique involving chloroform was performed (*Cole et al., 1978*). Microbial P was calculated as for $C_{mic}$ and $N_{mic}$ and converted using a $K_P$ value (the extractable part of microbial biomass P after fumigation) of 0.4 (*Lathja et al., 1999*). Microbial P was determined colorimetrically by the molybdate-ascorbic acid method using an Evolution 201 Thermo Scientific Inc. spectrophotometer (*Murphy & Riley, 1962*). Finally, $C_{mic}$, $N_{mic}$ and $P_{mic}$ values were normalized on a dry soil basis.

Because P is considered the most limited soil nutrient in the east-side of the CCB (Tapia-Torres et al., 2015), alkaline phosphatase activity was analyzed colorimetrically using $\rho$-nitrophenol ($\rho$NP) substrates, according to *Tabatabai & Bremner (1969)* and *Eivazi & Tabatabai (1977)*. For this analysis, 2 g of fresh soil and 30 ml of modified universal buffer (MUB) at pH 9 were used for the exoenzyme extraction. Three replicates and one control (sample without substrate) were prepared per sample. Three substrate controls (substrate without sample) were also included per assay. We centrifuged the tubes after the incubation period and then 750 µl of supernatant was diluted in 2 ml of deionized water and absorbance of $\rho$-nitrophenol ($\rho$NP) measured at 410 nm on an Evolution 201 Thermo Scientific Inc, spectrophotometer. Exoenzyme activities were expressed as micromoles of $\rho$NP formed per gram dry weight of soil per hour ($\mu$mol $\rho$NP [g SDW]$^{-1}$ h$^{-1}$). This value was standardized by $C_{mic}$ concentration for expression as a specific enzyme activity ($\mu$mol $\rho$NP [mg $C_{mic}$]$^{-1}$ h$^{-1}$).

### Molecular analyses

Total DNA was extracted using the hydroxyapatite spin-column method (*Purdy et al., 1996*). DNA molecular weight and quality were confirmed using agarose gel electrophoresis. The 16S rRNA gene was amplified from each sample using a polymerase chain reaction (PCR) with the universal primers F27 (5$'$AGAGTTTGATCMTGGCTCAG3$'$) and R1492 (5$'$GGTTACCTTGTTACGACTT3$'$). Three independent PCRs were performed for each sample. The PCR reactions were 50 µl in volume and contained 2µl of DNA, 1 µl PCR buffer 1$\times$ 0.5 mM $MgCl_2$, 0.2 mM dNTP mixture, 0.2 mM of each primer, 1 unit of platinum Taq DNA Polymerase High Fidelity (Invitrogen), 5% DMSO and 0.05 mg of BSA. The PCR was performed in a thermal cycler (MJ Research, Watertown, MA) under the following cycling program: initial denaturation step at 94 °C for 5 min, then 30 cycles at 94 °C for 1 min, 52 °C for 1 min,and 72 °C for 1 min 20 sec, with a final extension step at 72 °C for 30 min and storage at 4 °C. The three reactions were pooled and purified in a 1% agarose gel using the QIAquick gel extraction kit (Qiagen). The purified fragment was cloned into the vector PCR 2.1 and transformed into Escherichia coli following the manufacturer's instructions (Invitrogen). Only plasmids containing inserts were isolated for sequencing with the Montage Plasmid Miniprepkit (Millipore). The insertion within the plasmids was sequenced with the Sanger method using the vector-based primer 27F.

## Data analysis
### Stoichiometric homeostasis

The degree of community-level microbial C:N and C:P homeostasis ($H'$) by soil microorganisms was calculated with the formula proposed by *Sterner & Elser (2002)*:

$$H' = 1/m. \tag{1}$$

In Eq. (1), $m$ is the slope of $\log_e$ C:N$_R$ (Carbon and Nitrogen in the resources) versus $\log_e$ C:N$_B$ (Carbon and Nitrogen in the microbial biomass) or slope of $\log_e$ C:P$_R$ (Carbon and Phosphorus in the resources) versus $\log_e$ C:P$_B$ (Carbon and Phosphorus in the microbial biomass) scatterplot. $H' \gg 1$ represents strong stoichiometric homeostasis, while $H' \approx 1$ represents weak or no homeostasis (*Sterner & Elser, 2002*).

### Resistance and resilience index

Nutrient concentration and enzymatic activity data were both analyzed for resistance and resilience using the indices proposed by Orwin and Warlde (2004). The grassland site was considered as the control (C$_0$), the cultivated site as the disturbance (P$_0$) and the abandoned plot was used for measuring resilience 30 years after the cessation of agriculture management (P$_x$). Resistance (RS) was calculated as follows:

$$RS = 1 - ((2|D_0|)/(C_0 + |D_0|)). \tag{2}$$

In Eq. (2), C$_0$ represents the control soil and D$_0$ is the difference between C$_0$ and the disturbed plot (P$_0$). In addition, resilience (RL) was calculated as follows:

$$RL = ((2|D_0|)/(|D_0| + |D_X|)) - 1 (3). \tag{3}$$

In Eq. (3), D$_X$ is the difference between C$_0$ and P$_x$. Both indexes are bounded by $-1$ and $+1$, if the value is $-1$ means less resistance or resilience, while the $+1$ value means maximal resistance or resilience.

### Bioinformatics analysis

Sequencing quality evaluation as well as cloning vector removal were performed using the sorftware PHRED (*Ewing & Green, 1998*). For processing and classification of the sequence data, the open source software package Mothur (v 1.15.0; *Schloss et al., 2009*) was used. Sequences were screened for potential chimeric reads using Chimera.slayer (*Haas et al., 2011*) and the linked SILVA template database. High-quality sequences were compared against the SILVA database in order to obtain their taxonomic rank. A pairwise distance matrix was calculated across the non-redundant sequences, and reads were clustered into operational taxonomic units (OTUs) at 3% distance, using the furthest neighbor method (*Schloss & Handelsman, 2005*). In addition, the Simpson and Shannon (H) indices, Chao species richness estimator and rarefaction curves were estimated.

### Statistical analysis

One-way ANOVA was used to identify differences in nutrient concentrations and enzymatic activity between the sites of the agricultural gradient (grassland, cultivated field and abandoned field). Log-transformations were applied where the data deviated from normality. When ANOVA indicated a significant site effect, mean comparisons were performed with Tukey's multiple comparisons test (*Von Ende, 1993*).

Pearson correlations were used to explore relationships among soil parameters. Principal Components Analysis (PCA) was conducted in order to group soil samples with active

nutrients forms (dissolved, available and microbial) and enzymatic activity. Similarly, Canonical Analysis was conducted with soil nutrients (available, dissolved organic and pH) as the independent variables and nutrients within microbial biomass and phosphatase activity as dependent variables. All analyses were performed using R software 2.10.1 (*R Development Core Team, 2009*).

# RESULTS

## Soil nutrients

### Soil nutrients

The abandoned and cultivated plots had the highest and the lowest soil pH and soil electrical conductivity, respectively ($P < 0.0001$ and $P = 0.0002$ for pH and electrical conductivity, respectively; Table 1). Total organic C, N and P concentrations differed among management gradient plots. Total organic C was almost two times greater in the cultivated plot than in the other two plots ($P < 0.0001$; Table 1), whereas the cultivated and grassland plots presented the highest and the lowest N and P concentrations, respectively ($P < 0.001$ and $P < 0.0001$ for N and P, respectively; Table 1). As a consequence, the highest C:P and N:P ratios were in the grassland plot ($P < 0.0001$ for both C:P and N:P), while the C:N ratio did not differ among plots (Table 1).The cultivated plot presented higher DOC and DOP than the other two plots ($P < 0.0001$ and $P < 0.001$ for DOC and DOP, respectively), but DON presented no differences among plots (Table 1). Similarly, the cultivated plot presented a greater concentration of ammonium than the other two plots ($P < 0.0001$), but the highest values of nitrate and available P were in the abandoned and the grassland plots, respectively ($P < 0.0001$ for both $NO_3$ and available P; Table 1).

### Nutrients within microbial biomass

The cultivated plot had higher C and N concentrations within the microbial biomass ($P < 0.0001$ for both $C_{mic}$ and $N_{mic}$), but did not differ from the abandoned plot in terms of microbial P (Table 1). However, the grassland plot had higher $N_{mic}$ concentration than the abandoned plot and, consequently, the C:N and C:P ratios of the microbial biomass did not differ among plots, but the N:P ratio was highest in the cultivated plot ($P = 0.05$).

Using the equation for C:N and C:P homeostasis ($H'$), the soil microbial community did present a strong elemental homeostasis for phosphorus acquisition in the three sites ($H' = 6.25$, $9.35$ and $12.9$ respectively for cultivated, grassland and abandoned plots). For nitrogen acquisition, however, the microbial community of the cultivated soil presented a weak homeostasis ($H' = 0.63$), while the grassland (3.23) and abandoned plot (5.29) presented higher homeostasis.

### Enzymatic activity

The grassland soil had higher specific phosphatase activity than the other two managed plots ($P < 0.0001$; Table 1). The DOC correlated positively with DOP, ammonium, nutrients within microbial biomass and phosphanatase activity, while nitrate correlated negatively with available P and phosphanatase activity (Table 2). The first two principal components explained 74% of the total variance, in which 54% was explained by the first

**Table 1 Means (standard error) of available, dissolved, microbial forms of C, N and P and Specific phophonatase activity (SPA) of soil from an agricultural gradient at Cuatro Ciénegas Basin.** Values immediately followed by a different letter indicate that the means are significantly different ($P \leq 0.05$) among agricultural gradient plots.

| | Grassland | Cultivated plot | Abandoned plot |
|---|---|---|---|
| pH | 8.5 (0.03)[B] | 7.9 (0.04)[C] | 8.8 (0.04)[A] |
| EC (mS m$^{-1}$) | 8.7 (0.6)[B] | 3.4 (0.1)[C] | 15.6 (3.0)[A] |
| TOC (mg g$^{-1}$) | 5.97 (0.71)[B] | 21.50 (1.17)[A] | 9.54 (1.49)[B] |
| TN (mg g$^{-1}$) | 0.63 (0.06)[C] | 2.61 (0.07)[A] | 1.13 (0.05)[B] |
| TP (mg g$^{-1}$) | 0.094 (0.01)[C] | 0.768 (0.04)[A] | 0.53 (0.02)[B] |
| C:N | 9.3 (0.3) | 8.3 (0.6) | 8.3 (1.2) |
| C:P | 64 (5)[A] | 29 (2)[B] | 18 (3)[C] |
| N:P | 6.9 (0.5)[A] | 3.5 (0.2)[B] | 2.1 (0.1)[C] |
| DOC (µg g$^{-1}$) | 9 (2)[C] | 116 (9)[A] | 39 (7)[B] |
| DON (µg g$^{-1}$) | 7.7(0.8) | 6.6 (0.2) | 13.6 (3.5) |
| DOP (µg g$^{-1}$) | 1.1 (0.3)[B] | 14.6 (0.2)[A] | 2.1 (0.8)[B] |
| $NH_4^+$ (µg g$^{-1}$) | 1.64 (0.08)[B] | 3.51 (0.40)[A] | 1.55 (0.13)[B] |
| $NO_3^-$ (µg g$^{-1}$) | 0[C] | 4.91 (0.41)[B] | 18.16 (1.30)[A] |
| $HPO_4^-$ (µg g$^{-1}$) | 0.096 (0.015)[A] | 0.010 (0.002)[B] | 0.004 (0.001)[B] |
| $C_{mic}$ (µg g$^{-1}$) | 108 (12)[B] | 451 (68)[A] | 145 (29)[B] |
| $N_{mic}$ (µg g$^{-1}$) | 14 (1.3)[B] | 95 (23.6)[A] | 4 (1.0)[C] |
| $P_{mic}$ (µg g$^{-1}$) | 1.95 (0.41)[B] | 5.88 (1.21)[A] | 3.20 (0.48)[AB] |
| $C_{mic}$:$N_{mic}$ | 8.1 (0.9) | 9.00 (2.3) | 23 (6.9) |
| $C_{mic}$:$P_{mic}$ | 42 (9) | 99 (17) | 56 (13) |
| $N_{mic}$:$P_{mic}$ | 5.3 (1.1)[A] | 33.2 (16.4)[B] | 1.7 (0.3)[A] |
| SPA (µm mg$C_{mic}^{-1}$ h$^{-1}$) | 1.50 (0.44)[A] | 0.57 (0.08)[B] | 0.46 (0.27)[B] |

**Notes.**

EC, Electrical conductivity; TOC, totalorganic Carbon; TN, total Nitrogen; TP, total Phophorus; DOC, dissolved organic Carbon; DON, dissolved organic nitrogen; DOP, dissolved organic phosphorus; $NH_4^+$, ammonium; $NO_3^-$, nitrate; $HPO_4^-$, orthophosphate; $C_{mic}$, microbial carbon; $N_{mic}$, microbial nitrogen; $P_{mic}$, microbialphosphorus; SPA, specific phosphatase activity.

component. In the first component, the cultivated plot differed statistically to the other two non-cultivated plots, while all three plots were significantly different in the second component (Fig. 1). These results suggest that the difference between the cultivated plot and the other two plots explained 54% of the total variance in the soil nutrient dynamic. The dynamic forms of soil nutrients strongly correlated with nutrients within microbial biomass and phosphatase activity as determined by canonical analysis (Canonical $R = 0.98$, $P < 0.0001$). The eigenvalue of root 1 was 0.95 and pH and POD had the highest canonical weight in root 1.

## Soil resistance and resilience

In general, the soil variables analyzed showed low resistance to agricultural management, since the majority of the resistance values were negative or close to zero, with the exception of pH and DON (Table 3). Similarly, the soil variables also had low resilience, because none of the values was close to 1 (Table 3), which means that these soil variables were dissimilar

Hernández-Becerra et al. (2016), *PeerJ*, DOI 10.7717/peerj.2365

Peer J

**Table 2** Pearson correlation coefficients for available nutrients and nutrients within microbial biomass in soil from agricultural gradient at Cuatro Cienegas Basin.

| | pH | DOC | DON | DOP | $NH_4^*$ | $NO_3^-$ | $HPO_4^-$ | $C_{mic}$ | $N_{mic}$ | $P_{mic}$ | SPA |
|---|---|---|---|---|---|---|---|---|---|---|---|
| pH | 1 | | | | | | | | | | |
| DOC | −0.70* | 1 | | | | | | | | | |
| DON | 0.46* | −0.12 | 1 | | | | | | | | |
| DOP | −0.85* | 0.88* | −0.37* | 1 | | | | | | | |
| $NH_4^+$ | −0.68* | 0.65* | −0.23 | 0.72* | 1 | | | | | | |
| $NO_3^-$ | 0.59* | −0.01 | 0.46* | −0.19 | −0.21 | 1 | | | | | |
| $HPO_4^-$ | 0.09 | −0.51* | −0.17 | −0.44 | −0.27 | −0.61* | 1 | | | | |
| $C_{mic}$ | −0.68* | 0.79* | −0.24 | 0.74* | 0.70* | −0.09 | −0.32 | 1 | | | |
| $N_{mic}$ | −0.70* | 0.52* | −0.22 | 0.66* | 0.67* | −0.18 | −0.20 | 0.44* | 1 | | |
| $P_{mic}$ | −0.41* | 0.68* | −0.21 | 0.57* | 0.39* | −0.01 | −0.30 | 0.62* | 0.15 | 1 | |
| SPA | −0.88* | 0.65* | −0.40* | 0.84* | −0.76* | −0.52* | −0.11 | 0.64* | 0.62* | 0.30 | 1 |

**Notes.**

*Means significant correlation at $P \leq 0.05$.

DOC, dissolved organic Carbon; DON, dissolved organic nitrogen; DOP, dissolved organic phosphorus; NH4+, ammonium; $NO_3^-$, nitrate; $HPO_4^-$, orthophosphate; $C_{mic}$, microbial carbon; $N_{mic}$, microbial nitrogen; $P_{mic}$, microbial phosphorus; SPA, specific phosphatase activity.

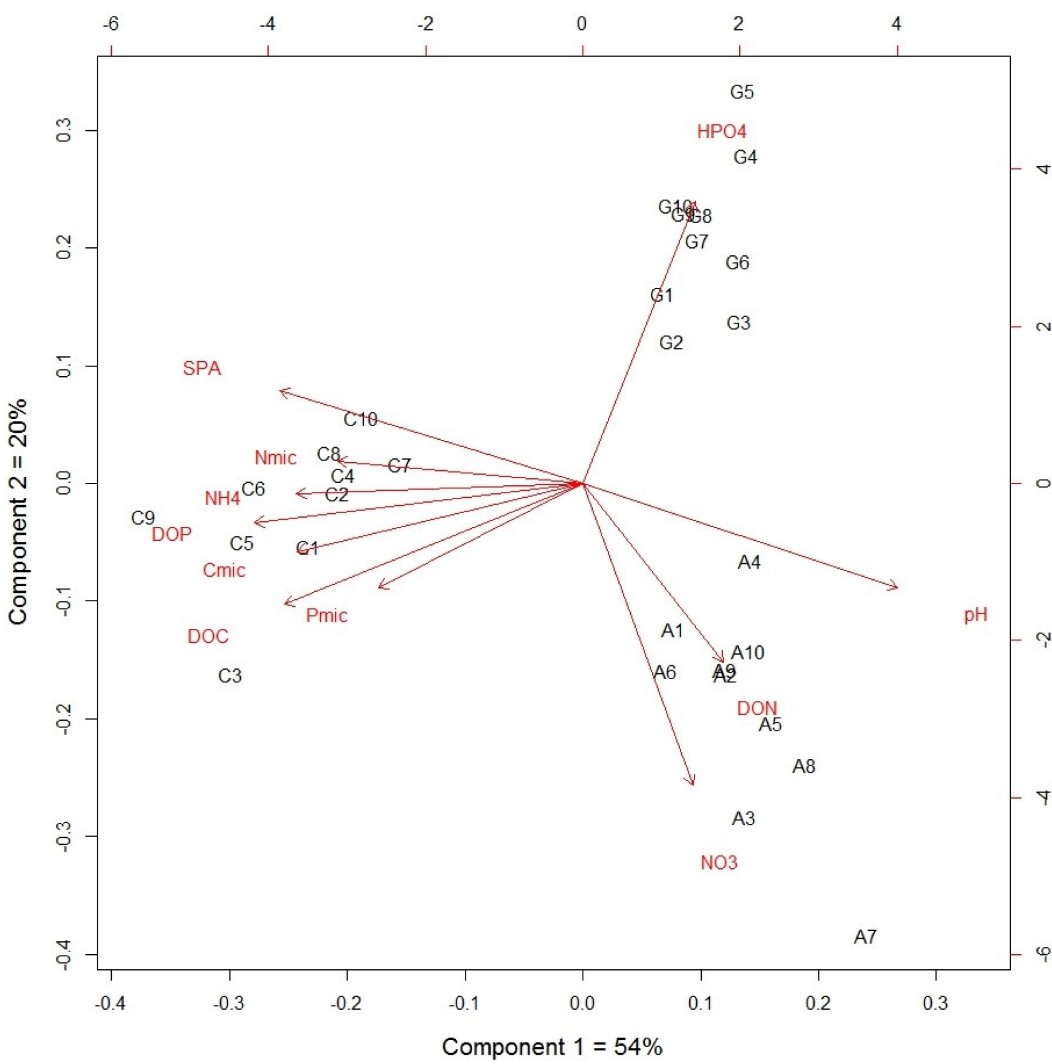

**Figure 1  Principal component analysis of dynamic nutrient forms from an agricultural gradient at Cuatro Cienegas Basin.**

to the grassland soil. However, the C and N concentrations within the microbial biomass, DOC and DOP were closer in value to 1 (above 0.5), suggesting that these soil variables were more resilient than the other soil variables analyzed (Table 3), although these values were insufficient to achieve recovery of these soil variables after 30 years.

## Soil bacteria composition
### *Composition of bacterial communities*

A total of 111 sequences were obtained for the grassland, 107 sequences for the cultivated plot and 93 sequences for the abandoned site. In the grassland, we obtained a clone library with 111 sequences, while the cultivated plot had 107 sequences and the abandoned plot had 93. In the grassland, the sequences were distributed among 12 phyla and 19 classes, while the cultivated plot sequences comprised 9 phyla and 14 classes, and those of the abandoned

**Table 3   Mean values (±standard error) of the resistance and resilience values of nutrient parameters from an agricultural gradient at Cuatro Cienegas Basin.**

| Variable | Resistance | Resilience |
|---|---|---|
| pH | 0.88 (±0.01) | 0.20 (±0.12) |
| DOC | −0.81 (±0.06) | 0.61 (±0.06) |
| DON | 0.54 (±0.08) | −0.28 (±0.18) |
| DOP | −0.84 (±0.04) | 0.81 (±0.06) |
| $NH_4^+$ | 0.04 (±0.15) | 0.42 (±0.16) |
| $NO_3^-$ | −1.00 (±0.00) | −0.57 (±0.03) |
| $HPO_4^+$ | 0.08 (±0.02) | −0.04 (±0.02) |
| $C_{mic}$ | −0.43 (±0.09) | 0.56 (±0.13) |
| $N_{mic}$ | −0.45 (±0.16) | 0.56 (±0.15) |
| $P_{mic}$ | −0.28 (±0.17) | 0.37 (±0.13) |
| SPA | −0.06 (0.10) | 0.25 (±0.12) |

**Notes.**

DOC, dissolved organic Carbon; DON, dissolved organic nitrogen; DOP, dissolved organic phosphorus; NH4+, ammonium; NO3−, nitrate; HPO4−, orthophosphate; $C_{mic}$, microbial carbon; $N_{mic}$, microbial nitrogen; $P_{mic}$, microbial phosphorus; SPA, specific phosphatase activity.

plot comprised 9 phyla and 12 classes. These results suggest that the bacterial community of the grassland soil was distributed in higher phyla than was the case in the other two managed plots. For example, Protobacteria was the more abundant bacteria phylum in the three plots, accounting for 50% of the results in the grassland and the abandoned plot, but representing only 35% in the cultivated plot (Fig. 2). Similarly, Actinobacteria was the second most dominant phylumin both the grassland and abandoned plot (20% and 21%, respectively), but only represented 15% in the cultivated plot. The two most important phototrophic phyla (Chloroflexi and Cyanobacteria) were not found in the cultivated plot, but Cyanobacteria was found in both the grassland soil and abandoned plot (Fig. 2).

### Diversity of bacterial communities

Rarefaction curve analysis showed that the cultivated plot had the richest bacterial community, followed by the abandoned plot and finally the grassland soil (Fig. 3). In addition, the cultivated plot had the highest expected OTUs by the Chao analyses (659), while the abandoned plot had the lowest expected value of OTUs (179). The latter plot also had the lowest values of Simpson and Shannon indices ($D = 0.025$ and $H = 3.8$, respectively), suggesting that the bacterial community of the abandoned plot was dominated by fewer OTUs in comparison with the bacterial communities in the cultivated plot and the grassland soil ($D = 0.04$, $H = 4.4$ and $D = 0.013$, $H = 4.2$; respectively).

From the total of 307 sequences obtained for all sites, 223 OTUs were recognized at 97% of similitude. The cultivated plot again had the highest number of OTUs (92), followed by grassland (84 OTUs) and finally the abandoned plot with the lowest number of OTUs (59). The three sites shared four OTUs corresponding to the Proteobacteria (Rhizobiales, Pseudomonadales, Burkholderiales and Xanthomonadales). The abandoned plot shared two OTUs with the other sites, but there were no OTUs shared between the grassland

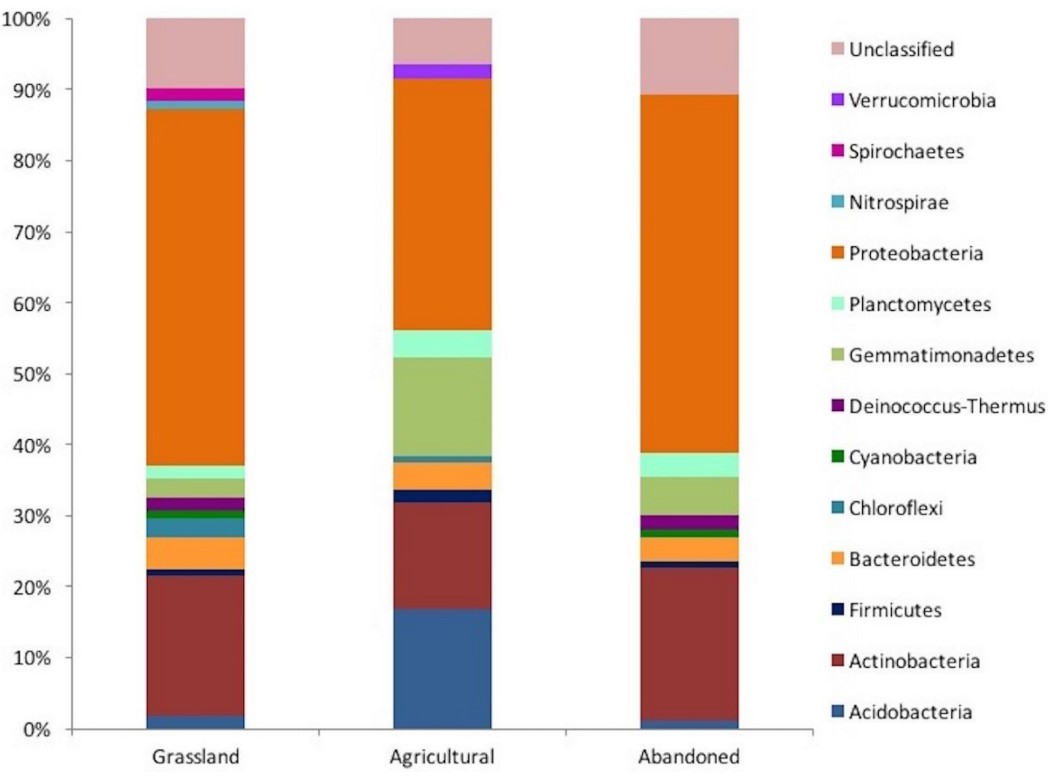

**Figure 2** Taxonomic distribution of the 16 rRNA gene sequences obtained from clone libraries of an agricultural gradient at Cuatro Cienegas Basin.

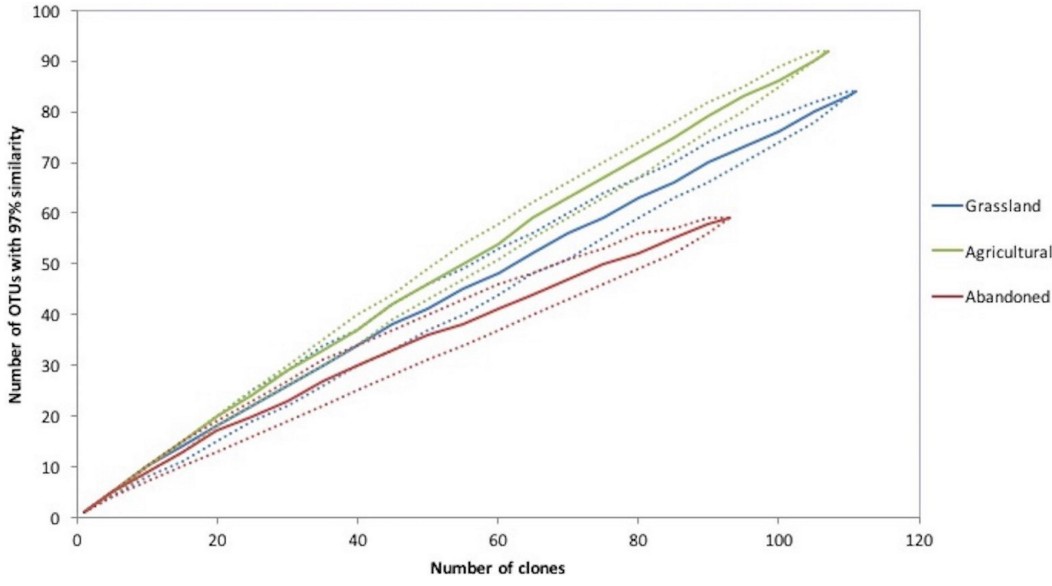

**Figure 3** Rarefaction curves of an agricultural gradient at Cuatro Cienegas Basin. OTUs were determined at 97% sequence identity.

and the cultivated plot. Finally, the grassland soil and abandoned plot presented higher similitude between them relative to the cultivated plot, using the 16S rRNA community composition at 97% similarity based on the Bray-Curtis algorithm.

## DISCUSSION

### Soil nutrient dynamics

In the Cuatro Cienegas basin (CCB), alfalfa production by flooding the fields threatens the wetlands sustainability and contributes to the degradation of soil and vegetation system. The results showed that the cultivated plot presented a lower soil pH than the other two sites, which could be associated with the fertilization and continuous irrigation, it has been reported in other agriculture sites (*Moore, Klose & Tabatabai, 2000*; *Raiesi, 2004*). Soil N fertilization mainly with ammonium, as it is applied to the site of the present study, promotes nitrification by releasing $H^+$ ions into the soil solution (*Moore, Klose & Tabatabai, 2000*), while continuous irrigation increases the leaching of salt through the soil profile (*Raiesi, 2004*). However, the cultivated plot presented higher concentrations of total C, N and P than the other two plots. These increases are caused by fertilization and by the particular crop under cultivation, with the latter mainly affecting the SOC concentration. Perennial legumes, such as alfalfa, promote higher SOC accumulation in comparison with the annual crops since they feature high root biomass production and require low soil tillage (*Franzluebbers, 2009*; *Sainju & Lenssen, 2011*; *Bell et al., 2012*; *Yang et al., 2013*). Furthermore, the alfalfa plot had a greater availability of dissolved organic carbon (DOC), which could be explained by higher organic matter input and soil water availability. These conditions promoted depolymerization of organic molecules and mineralization of organic nutrients mediated by the activity of heterotrophic microorganisms (*Wardle, 1992*; *Vineela et al., 2008*). Associated with this higher activity of heterotrophic microorganisms, organic N is mainly released as $NH_4^+$ and then immobilized within microbial biomass, as suggested by the $NH_4^+$ and $N_{mic}$ values of the cultivated plot. All of these results suggest that the cultivated plot presented higher soil nutrient transformations, mainly of N, promoted by the availability of water and nutrient fertilization, and thus the soil nutrient dynamics of this plot differ from the plots without management, as suggested by the results of the PCA. In contrast, the low amount of soil organic matter in the native grassland is consequence of low availability of soil water in the east-side of CCB (*Tapia-Torres et al., 2015a*). The low water availability reduces plant productivity and in consequence there is a lower input of organic matter input to the soil, as *Tapia-Torres et al. (2015b)* reported for soils under desert scrub within CCB. Consequently, the activity of microbial populations is constrained by low availability of organic carbon (*Wardle, 1992*).

The nutrients within microbial biomass and phosphatase activity are strongly affected by the dynamics of soil nutrients as Canonical Analysis confirmed. While the cultivated plot presented higher nutrient concentrations within microbial biomass than the other two plots, microbial C:N and C:P did not differ among plots. These results suggest that the soil microbial community had different strategies for nutrient acquisition in order to equilibrate nutrient stoichiometry (*Sterner & Elser, 2002*). The soil microbial communities

of the plots showed elemental homeostasis, with the exception of the cultivated soil, in which N acquisition showed weak homeostasis, probably in response to the constant fertilization with ammonium. *Tapia-Torres et al. (2015a)* also reported a strong N and P homeostasis for two native grasslands within the CCB. These results suggest that soil microbial communities adopt different strategies for nutrient acquisition, including the production of eco-enzymes which clave the organic molecules for microbial assimilation (*Waring, Weintraub & Sinsabaugh, 2014*). Phosphatase is the main eco-enzyme that mineralizes organic P molecules (*Tabatabai & Bremner, 1969*). In our study site, the native grassland had higher specific phosphatase activity, indicating that members of the soil microbial community in this plot invest more in production of this enzyme than in growth, which suggests that this microbial community is co-limited by C and P as reported before for the same study site by *Tapia-Torres et al. (2015a)*. Moreover, the microbial C:N:P ratio of the cultivated plot (99:33:1) is wider than that proposed by *Cleveland & Liptzin (2007)* for different terrestrial ecosystems (60:7:1), while the non-managed plots are closer to this ratio (42:5:1 and 56:2:1 for the natural grassland and abandoned plot, respectively). These results suggest that the agricultural management strongly disrupts soil microbial activity and its homeostasis.

As expected, the sites under no current management were limited by water and DOC. At the abandoned site, these conditions promoted the nitrification process, which is mediated by autotrophic microorganisms that can use $NH_4^+$ as their energy source (*Hart et al., 1994*; *Chen & Stark, 2000*). The microbial N immobilization process was favored in the native grassland; this process promotes N conservation within the ecosystem, as previously reported for native grassland in the CCB (*Tapia-Torres et al., 2015b*).

## Soil bacteria composition

The agricultural land-use change affected the soil bacteria composition. Agricultural management increased the numbers of OTUs and diversity indices associated with higher availability of soil water and energy for microbial activity. Such increases due to agriculture activity have been reported for other desert sites (*Köberl et al., 2011*; *Wang et al., 2012*). However, the abandoned plot had lower OTUs and diversity indexes in comparison with the other two plots, probably associated with more stressful soil conditions (i.e., higher salinity, lower water and nutrient availability) as reported by *Keshri, Mody & Jha (2013)* for desert soils.

The two dominant phyla from the three plots analyzed were Proteobacteria and Actinobacteria, which are both very common in agricultural (*Buckle & Schmidt, 2003*; *Chaudhry et al., 2012*) and desert (*Chanal et al., 2006*; *López-Lozano et al., 2012*) soils. However, their relative proportion differed among plots, especially in the case of the cultivated plot. Moreover, the two most important phototrophic phyla (Chloroflexi and Cyanobacteria) were not found in the cultivated plot, where N input and soil disruption selected against their presence. As expected, Cyanobacteria were present in both the grassland soil and the abandoned plot, forming a desert crust (*Li et al., 2012*). In contrast, the Acidobacteria were more abundant in the cultivated plot (*ca.* 18%), while in the non-cultivated plot had decreased to 2%. This phylum is associated with pH neutral or acid soils,

such as the soils of the cultivated plot. The results suggest that agricultural management has a strong effect on soil bacterial composition, because the agricultural plot shared lower OTUs (only 4) with the plot under no management. Furthermore, according to the Bray-Curtis algorithm, the grassland soil and the abandoned plot had a higher similitude between them relative to the cultivated plot. For example, in both the native grassland and the abandoned plots, some extremophile OTUs were present, e.g., that associated with the phylum Deinococcus-Thermus, which is adapted to stressful soil conditions such as salinity, high temperatures, aridity, etc., (*Nienow, 2009*); however, these OTUs were not presented in the cultivated plot. These results suggest that some OTUs recover after abandonment of agricultural management, although the soil bacteria community is not yet similar to that in the native grassland even after more than 30 years since abandonment. One study has reported similar soil bacteria in native vegetation and sites abandoned for over 45 years in agro-ecosystems of Michigan State (*Buckle & Schmidt, 2003*).

In soil microbial communities, microfungi are an important and diverse component of microbial diversity, representing a large proportion of microbial diversity in soils (*Fierer, Bradford & Jackson, 2007*). These microorganisms play an immense role in regulating energy and nutrient fluxes through natural ecosystems, via their involvement in soil development, decomposition and uptake of nutrients by plants (*Dighton, 1997*) mainly phosphate uptake. Future studies should be aimed at understanding the role of microfungi in soil nutrient cycling in this ecosystem. However, tagging of ITS markers for soil fungi in CCB have been challenging, so there is still further research needed in this field.

## Soil resistance and resilience

All of the variables evaluated presented low resistance and resilience, suggesting that the native grassland soil may be very vulnerable to agricultural transformation. The resilience of soil is determined by its intrinsic characteristics, as well as by prevailing climatic conditions (*Blanco-Canqui & Lal, 2010*). For instance, soil with high organic matter content is more resilient, since organic compounds represent important reservoirs of energy and nutrients for both the soil microbial community and plants (*Bronick & Lal, 2005*). In addition, ecosystems in humid climates are also more resilient than arid ecosystems because they are not constrained by water availability. For example, the wet tropical ecosystem requires less than 10 years for recovery of its vegetal community following perturbation, while the desert ecosystem requires at least 40 years (*Lesschen et al., 2008*; *Wang et al., 2011*). Our results suggest that the native grassland presents slow recovery and this characteristic is critical for the design of alternative agricultural management, as well as appropriate strategies for soil reclamation. This is important because the rate of soil degradation is faster than that of soil restoration, which acts to increase the area of degraded lands in these arid ecosystems.

The design of soil restoration practices is critical for CCB, because the ecosystems within CCB are very vulnerable to the disruption of nutrient dynamics, and the native species have low competition capacity against invasive species under higher availability of resources (*Souza et al., 2006*). This situation is critical for the soils of CCB, because they contain a high diversity of native species that can face up the scarcity of nutrients, mainly P (*Tapia-Torres et al., 2016*). The organic agriculture with low pesticide inputs and the use

of native microbial strains with different capabilities to use, transform and recycle the soil nutrients (i.e., phosphorus solubilizing bacteria) could be the best solution for agriculture in this particular and highly diverse important ecosystem. These agricultural practices not only will allow the maintenance of soil microbial biodiversity but also will contribute to the soil conservation. Therefore, ensuring long-term availability and accessibility to healthy soil, mainly for food security is a global challenge.

## CONCLUSIONS

Our results suggest that land-use change transforming native grassland into agricultural land induces drastic modifications in the soil nutrient dynamics as well as in the bacterial community. However, with the suspension of agricultural practices, some soil characteristics tend to slowly recover their natural state.

## ACKNOWLEDGEMENTS

We are thankful to Rodrigo Velázquez-Durán for assisting with chemical analysis and to Alberto Valencia for assisting data analyses. We also thank Oscar Sánchez Liceaga, Héctor Castillo González, the personnel of APFF Cuatro Cienegas (CONANP) and the people in charge of Rancho Pozas Azules (PRONATURA) for permission to collect soil samples on their respective properties.

### Funding

This work was financed by the National Autonomous University of Mexico (PAPIIT DGAPA-UNAM grant to FGO: Análisis de la vulnerabilidad de la dinámica de nutrientes en un ecosistema árido de México, IN204013), as well as a grant from the alianze WWF-FCS to VS. The funders had no role in study design, data collection and analysis, decision to publish, or preparation of the manuscript.

### Grant Disclosures

The following grant information was disclosed by the authors:
National Autonomous University of Mexico.

### Competing Interests

Valeria Souza is an Academic Editor for PeerJ.

### Author Contributions

- Natali Hernández-Becerra conceived and designed the experiments, performed the experiments, wrote the paper, prepared figures and/or tables.
- Yunuen Tapia-Torres conceived and designed the experiments, analyzed the data, wrote the paper, prepared figures and/or tables.
- Ofelia Beltrán-Paz and Jazmín Blaz performed the experiments, reviewed drafts of the paper.

- Valeria Souza analyzed the data, contributed reagents/materials/analysis tools, reviewed drafts of the paper.
- Felipe García-Oliva conceived and designed the experiments, analyzed the data, wrote the paper.

### Field Study Permissions

The following information was supplied relating to field study approvals (i.e., approving body and any reference numbers):

Oscar Sánchez Liceaga, Héctor Castillo González, the personnel of APFF Cuatro Cienegas (CONANP) and the people in charge of Rancho Pozas Azules (PRONATURA) gave us permission to collect soil samples on their respective properties.

The owners gave us oral permission to collect soil samples.

### Data Availability

The raw data has been supplied as Data S1.

### Supplemental Information

Supplemental information for this article can be found online at http://dx.doi.org/10.7717/peerj.2365#supplemental-information.

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
