# Peer review of "Agricultural land-use change in a Mexican oligotrophic desert depletes ecosystem stability"

_PeerJ, doi:10.7717/peerj.2365_

## Round 0.1 · original submission · Minor Revisions

Besides the remarks and suggestions of each of the 3 reviewers, specific attention is required with regard to i) the possible role of soil fungi in the study, and ii) executing a canonical analysis with the independent variables (soil nutrients) and the dependent variables (microorganisms nutrients, and diversity of microorganisms) to further and/or better underpin observed results.

·

Basic reporting

In my opinion the article is writing in a good form and it is into the topic of the Journal. I am not native English but the redaction is clear. This study addresses is an interesting issue related to the influences of the agricultural management on soil bacterial community in arid environment. The theoretical basis of this paper is quite solid.

Figure 1 could be improved and include in PCA the distribution of soil properties analysed, this will help to understand the response of soil to different use.

In my opinion the table 2 must be removed because there is not significant correlations between different parameters, if any correlation is important is better include the comment in text.

Experimental design

The experimental design is good, but I miss some important data as the EC of soil, also data about the quality of water. The author explain that the irrigation increase the leaching salt in soil profile but does not provide any data about.

Validity of the findings

The study is focuses on the influence of land use on some nutrients dynamics and bacterial community, in my opinion should be very interesting include also the effect on fungi community because these microorganism are key in many properties of soil, please justify why it has not been analysed this community.

Additional comments

The discussion is adequate with the aim of study but so much descriptive. Please try to improve the explanation of findings of the low content of OC and nutrients in the native soils and also the low carbon biomass and microbial activity.
I suggets for review:

http://journals.plos.org/plosone/article?id=10.1371/journal.pone.0080522

Reviewer 2 ·

Basic reporting

In this manuscript, the effect of different land managements is estimated in term of soil nutrients, enzyme activity, and structure of microbial community. The topic is a very interesting because the Mexican desert region is poor investigated in this aspect until now. Research question well defined, structure of MS conforms to PeerJ standard, literature well referenced. Tables and figures are high quality and well described.

Experimental design

Experimental design is carefully thought-out. Methods and approaches are modern and described with sufficient details. Statistical analysis is correct and completely corresponds to the goals of present study.

Validity of the findings

The results obtained are clear, discussion is quite exhaustive.
The impact and relevance of this study should be explained more specifically.

Additional comments

Some corrections are required before publication in Peer Journal

Specific comments:
Lines 83-84: provide, please, the information on areas (ratio) of native grasslands, abandoned and agricultural lands in desert Mexico

Lines124-125: It would be nice to provide more information on fertilization of agricultural plots (rate and kind of fertilizers) and what is period of alfa-alfa cultivation here. What crops rotation is typical for this region.

Lines 150-210: it is advised to divide the section “ Soil nutrient and enzymatic analyses” into several subsections: soil nutrient; C, N, and P in microbial biomass; enzymatic analyses
It would be nice to provide the explanation why you choose the phosphatase activity (only) to estimate the enzymatic activity of soils.

Line 237: what the negative values of homeostasis ( H´<0) mean. Explain, please.

Line 238-248: Provide, please, the additional information what show the different intervals of Resistance and resilience indexes to understand better their ecological meaning.

Line 294: – Here, it is better to start a new paragraph (about correlation, and other data analyses)

Table 2: Phospo means SPA (as Table 1), is not it? Use, please, the same terminology for all tables. For tables 2 and 3, you should note, that you use the abbreviations similar to table 1.

Check carefully your English, especially the punctuation and articles.

Reviewer 3 ·

Basic reporting

Clear, unambiguous, professional English language used throughout. Yes it is a good English language
Abstract is well written, it is clear
Key words: I suggest to put the scientific name of the Alfalfa

-Intro & background to show context. It is well done,
Line 95, please use the scientific name of Alfalfa, Li

-Literature well referenced & relevant. Yes

-Structure conforms to PeerJ standard, Yes

-discipline norm, or improved for clarity.

-Figures are relevant, high quality, well labelled & described. Figures are good, but I suggest to add in the Principal components analysis, also the diagram of the variables: nutrients dissolved, active nutrients, available nutrients and microbial nutrients, we can see very well the sites, but it is important also to see the position of the variables over the axes, and to see which variables are correlated of not.

Raw data supplied (See PeerJ policy). Not, I didn’t find it.

Experimental design

Original primary research within Scope of the journal. Yes very good

Research question well defined, relevant & meaningful. It is stated how research fills an identified knowledge gap. Yes well done

Rigorous investigation performed to a high technical & ethical standard. Yes

Methods described with sufficient detail & information to replicate. Yes, but I suggest to explain better the equation 1, and in Equation 3 it is important to describe Px. Please inform carefully of the type of agriculture that takes place there, the amount and frequency of the chemical inputs used in the agricultural system.

Validity of the findings

Impact and novelty not assessed. It is a very interesting study, and the results are well presented, but even though they are well organized, I suggest inside the nutrients section a subsection informing about the nutrients measured in the organisms, in order to clarify how are the conditions, otherwise is confusing to know which nutrients belong to the soil and which nutrients to the organisms, I find interesting the PCA, but, I suggest also to add the diagram of the variables, in order to see how are the variables over the axis. All the abbreviations used in the table 1 must be also described at the beginning of the results section, in Table 2 is important to add the description of the abbreviations. Maybe would be interesting to see a canonical analysis, with all the variables, the independent variables (soil nutrients) and the dependent variables (microorganisms nutrients, and diversity of microorganisms), then I encourage the authors to do also a canonical analysis.

Additional comments

It is a very interesting study, the methods are well described and the results are well described, but I encourage you to do a canonical analysis (please see file), I also encourage you to discuss more which is the relevance of the area, why CCB can be very vulnerable, why it has been protected, why the agriculture or the type of agriculture can be nocive for the system, explain more, and give recommendations.

Annotated reviews are not available for download in order to protect the identity of reviewers who chose to remain anonymous.

---

## Round 0.2 · accepted · Accept

The paper has been improved successfully by the authors, addressing all remarks and suggestions of the respective reviewers.

Congratulations with this achievement!